# Revelations of service-learning project: Multiple perspectives of college students' reflection

**Ting-Hua Lin** *

Department of Early Childhood Care and Education, Cheng Shiu University, Kaohsiung, Taiwan

* k0367@gcloud.csu.edu.tw

## Abstract

This study explored the reflections of college students at different stages of their service provided to preschoolers and proposed recommendations on service-learning curricula according to the research findings. Thirty-six college juniors enrolled in an academic service-learning course were asked to complete a service-learning project, to undertake service activities in the form of eight storytelling sessions at preschools. This storytelling plan was 1.5-hour-long each, twice per month, for 16-weeks. Qualitative case study approach was employed in this study. Data consisted of documentation of the participants' reflections, focus group interviews, and research logs. The thematic analysis was employed to analyze data and data were analyzed using the constant comparative technique. The study revealed that the contents of participants' reflections took on a qualitative change as their service experience accumulated. Before their service experience, their reflections focused on personal expectations. This shifted professional learning and personal growth during the service experience. After their service experiences, their reflections were extended to include social concern. Also, the participants were inspired to actively acquire professional knowledge from their service experiences. Various reflection activities triggered different levels and functions of reflection by the participants. Writing activities caused the participants to focus on personal reflection, providing the opportunity for them to examine and adjust their ideas; group discussion activities allowed the participants to focus on group dialogue, with which they achieved learning and gained identity. Finally, students experienced emotional highs and lows during their experience of service, describing tension and fear initially, then joy or disquietude subsequently. Recommendations based on the research results were provided to facilitate planning of the service-learning curriculum by college educators.

## 1. Introduction

In recent years, the integration of service-learning (SL) into the professional education curriculum has become a trend in higher education institutions [1–3], as college educators experiment with innovative teaching strategies [4–9]. The educational effectiveness of SL has led to its wider adoption around the world [10]. The positive effect of SL on the student learning

**Data Availability Statement:** All relevant data are within the paper.

**Funding:** The authors received no specific funding for this work.

**Competing interests:** The authors have declared that no competing interests exist.

experience can be attributed to the activities designed for students to reflect on their service [11–15]. Reflection is an increasingly critical component in the higher education sector, because it is a type of learning based on experience that encourages students to create connections between theory, practical knowledge, and personal experience [2,10,16].

McPherson observed that the effects of reflective activities involved in SL included effective problem-solving strategies; lifelong learning skills; greater ability to create and change; the ability to assess the root causes of complex issues with more intellectual sophistication, refined perspectives on service activities and the entire scheme, the ability to overcome personal stress and anxiety, enhanced group cohesion and trust, leadership, citizen rights, critical thinking skills, and problem-solving skills [17]. Researches considered the continuity of the reflection to be the focus of service-learning [17,18]. Reflection is a continuous process that traverses the entirety of the SL curriculum and is not only implemented at a specific stage of service-learning [19]. Pre-service reflection is to clarify students' expectations and preparations. In-service reflection is the ability of students to learn and practice and the difference between adjustment expectations and real experience. Post-service reflection is the student's willingness to review the entire service process and guide students' future services [11,12].

Previous studies of service-learning have shown a positive impact on student learning outcomes [2,3,5–7,20–22]. The quality of reflection activities affects learning outcomes related to service-learning [23]. In particular, the quality of reflection focuses on the structure and clarity of reflection practice [24,25]. By means of group discussion or written reflection activities, students can share their learning experience together [5,14], make their tacit knowledge explicit [13], and use new cognition in the next service-learning [11]. According to Jacoby, SL can promote this process through providing challenges and the support to overcome these challenges. SL can affect students' cognitive, psychosocial and social identity development [12]. With the accumulation of SL experience, students may change their reflections in terms of content and perspective as well as improve their learning and change [11]. For example, Chien [26] found that students' interpersonal interaction and self-perception have obvious progress in-service reflection and post-service reflection.

Accordingly, this study aimed to answer questions regarding the contents of students' reflections at various stages of the academic SL curriculum. How do reflection contents differ at various stages? What revelations can the analysis of reflection contents yield for service-learning? These research questions may provide insight for college educators to understand the reflection perspectives of students in the SL process and may serve as a reference for the design of SL reflection activities.

## 2. Literature review

### 2.1 Service-learning synopsis

Institutions of higher education across the world have begun actively promoting the SL curriculum in light of the striking results that the United States has achieved in implementing service-learning [27]. Kendall [28] remarked that service-learning has been perceived as a scheme, a philosophy, or pedagogy. SL is a type of pedagogy that achieves an instruction goal, namely connecting community services with educational institutions [6,23,29,30]. It is currently considered a high-impact educational practice [31].

SL is also regarded as a systematic, community demand-driven service activity in which students can participate to achieve a deeper understanding of the curriculum content and an enhanced sense of civic responsibility through reflection [32]. Hsu [33] took SL as an experimental education teaching method as well as a reciprocal philosophy. It can be purposely designed to function with the principles of reciprocity and reflection and with emphasis on

service and learning objectives, facilitating the service providers' and service recipients' attainment of their respective goals.

SL places a premium on experiential learning and reflection [14]. Through the experiences of connecting actions with knowledge and service processes, it extends reflection for generating new knowledge and solutions to problems [6]. In the repetitive and cyclic processes of service-learning, students combine theory and practice and augment their awareness of social issues through. Finally, students internalize the process to create personal habits and attitudes that are beneficial to themselves and to society. SL has the following key elements: (1) benefitting the community through academic, social, and personal skills; (2) making decisions in real-world, non-hypothetical contexts; (3) promoting personal growth, peer support, and civic engagement; (4) accumulating successful experiences; (5) attaining more profound knowledge of the self, the community, and society; (6)deep reflection; (7)broad dissemination of insights; (8) reciprocity between students and partners; and (9) developing leadership, problem-solving, and teamwork skills in the course of serving others [34,35].

In summary, SL integrates educational institutions and communities and serves as implementation of the philosophies and curriculum of the flipped education. Instructors designing courses that seek to establish a learning field encompassing the campus and the community to extend student learning experience from the classroom to the context of social service [2], turning school subjects into service elements. In the processes of actual experience and reflection, students achieved learning effectiveness in dimensions such as professional learning, personal growth, and social responsibility [12].

## 2.2 Theoretical basis of service-learning reflection

SL experience may have transformational impact for students [18], it includes the cycle of service action, accompanied by opportunities for reflection [13]. Reflection requires practical experience of learning; specifically, an individual identifies problems in their experiences and then engages in the reflection process to seek solutions [14]. Reflection is how learning takes place, and experience does not necessarily generate learning and comprehension [19,36]. Reflection must continue in the SL curriculum. Continue means the students can achieve deep learning for the pre-experience, in-experience and post-experience [12]. Ash [37] stated that participant reflection process does not happen automatically, but must be designed carefully and intentionally.

In Taiwan, the "University Service-Learning Program" promulgated by the Ministry of Education in 2007, focuses on service-learning integrated with courses, and aims to effectively combine "community service" with "academic courses". Therefore, the application of service-learning courses is widely promoted by university teachers. For teacher educators, service-learning is regarded as a teacher education programs [3,6,20]. The service targets of these programs range from preschoolers to high school students. Among them, service-learning for young children is combined with professional courses [5,21,22]. These service-learning setting include preschools and zoos [5,7,21,22]. The service-learning content includes planning the preschool learning area and leading children to activities in the learning area [21], as well as learning activities that design activities and lead children to explore nature [5,22].

The theoretical basis of reflection on service-learning is as follows. (a) Dewey's experiential learning model: Dewey regarded reflection as the reconstruction and reorganization of experience, a unique problem-solving approach, and a means of learning with which learning outcomes can be achieved [38]. Therefore, learning is a process of generating experiences from reflection in challenging circumstances and converting such experiences into knowledge. (b) Kolb's experiential learning cycle: Individuals perform reflective learning by establishing a

learning cycle of several stages, namely concrete experience, reflective observation, abstract conceptualization, and active experimentation [39]. This theory supports the argument that SL is a type of pedagogy and holds that reflection is a critical factor in theoretical and practical SL experiences. (c) The accordion effect of Sheckley [40]: The two authors considered the processes of reflective judgment and reflective action to be the core of experiential learning. When students' new experiences and expectations differ, students rethink, reconceptualize, and even change their perspectives to learn from their own experiences despite setbacks and pressures. (d) Social identity development theory: This theory holds that SL participants can only produce cognitive, sentimental, and behavioral changes when they perceive and reflect upon the attitudes of other groups in their own experiences [19]. The aforementioned perspectives demonstrate that student experience and reflection during the SL process interact and produce learning effects; therefore, the process enables students to acquire new concepts and promotes the development of holistic education.

Practical qualities of SL reflection include the following: (a) Connection: this quality emphasizes the inseparability of classroom from community learning, sentimental learning from cognitive learning, and experience from practice. (b) Continuity: SL curricula require continuous arrangement of reflective activities before, during, and after individuals' service. (c) Context: The service experience and the choice of institutions must be relevant to the students' learning content to provide students with an integrative learning and service experience. (d) Challenge: Learning occurs when new experience and information conflict with preconceived concepts, using confusion appearing in the thinking process to prompt growth. (e) Coaching: Challenging preconceived concepts is at the core of growth. Students are prone to experience frustration and confusion in the process of developing new perspectives. Instructors must offer an appropriate degree of appreciation and support and must guide students to accommodate new perspectives to process their experience and observation in relation to their current understanding and develop alternative interpretations [17,18,32,41].

The function of SL reflection includes the following: (a) Motivate students to seek focal points among the various stimuli during their service and increase their engagement in the service. (b) Assist students in coping personal issues regarding adaptation and obtaining emotional support that permits them to remain committed to the continuity of service. (c) Enhance students' group discussion and sharing of their personal reflection during service to integrate new and old experiences, thus attaining a holistic learning process. (d) Disseminate students' post-service reflection reports through various means to affect policy and to prompt others to participate in service-learning [15].

Sherman [42] proposed several structured reflection questions for the SL curriculum: (a) What is the relationship between activities undertaken during service and the concepts learned in the classroom? (b) What unique concepts, theories, or skills learned in the curriculum are relevant to real-world situations? (c) Why are such services provided according to the demands of community partners? (d) Who benefits from the service? (e) What underlying issues exist? (f) How do services differ? Who is the target? (g) What did I learn from the course contents, myself, and the community? These questions can be integrated into the reflection activities as appropriate to guide and assist students in focusing on key points of the activities and navigate students during the process.

Reflection is established to be a fundamental factor for SL curricula [11,12]. Students participate in the structured reflection activities designed by the educators, in which students' actions during service and their learning experiences are connected in an effective manner, generating new cognition, sentiments, and skills in relation to oneself, the beneficiaries of the service, and society [23]. Activities enabling reflection must remain continuous in the SL process to achieve professional learning and the fostering of civic literacy in reflection-in-action.

## 2.3 Learning outcomes of reflection in service-learning curriculum

Service-learning provides pre-service teachers with more field experience to help them understand how children's family and community life affects their behavior in school. Pre-service teachers integrate experience into their identities as teachers by reflecting on the service-learning process [43]. Structured Reflection indicates the success or failure of a SL curriculum [11]. Relevant research has found reflection content to be predominantly concerned with student learning outcomes, including professional learning, individual growth, and social concern. Each is described in the following.

a. Professional learning: Improve student understanding and practical capabilities in relation to adaptive education [5]; enhance students' pedagogical knowledge, capabilities of planning and implementing collaborative curriculum, skills of designing crossdisciplinary pedagogical activities, and professional attitudes [7]; help students develop curriculum-design and pedagogical capabilities, gain greater understanding of pedagogical events, maintain favorable instructor–student relationships, develop counseling abilities, enhance management skills, improve interaction skills with instructors, and foster ethical professional values and attitudes [3]; enhance the self-efficacy of student' teaching technology, the belief in the integration of teaching technology, and understand the role of teaching technology in education [44]; development of professional identity of pre-service teachers [45]; and developing classroom and behavior management skills [46].

b. Individual Growth: Boost students' self-worth [47]; promote personal growth and career exploration [20,48]; and guide students to perceive positively, identify personal weaknesses, hone communication skills and problem solving abilities, stimulate ideas through peer observation and collaboration, provide collective emotional support, learn to care for others, explore their career, stimulate service dedication, improvise actions according to immediate circumstances, and permit self-reflection [7,21].

c. Social Concern: Improve students' sense of social responsibility and civic engagement [20,48,49]; nurture students' humanistic and social concerns and allow students to experience ethnic differences and care for the disadvantaged [3,48]; make students aware of factors affecting preschool education quality [21]; provide students with information regarding social attitudes, leadership skill development, and opinions in the community [50]; enhance students' understanding of the importance of community involvement [51]; and how they can contribute to the community [52].

Reflection is a critical factor in students' learning outcomes and also an essential consideration for instructors when designing SL curricula [11]. Only through appropriate reflection activity planning can SL curriculum objectives be achieved. Students' reflections reflected the outcome of the SL curriculum, including professional learning, individual growth, and social concern. The aforementioned account provides a referential basis for this study's analysis of students' reflection content.

## 3. Methodology

### 3.1 Design and implementation of reflection activities on service experiences

Reflection activities in this research describe activities designed for this study to implement the academic curriculum integrated with SL. The service-learning program in this study refers to participants who completed the Preschool Classroom Management course used the knowledge they obtain from the course, including ground rules, classroom discipline, time management

Table 1. Preschool storytelling service-learning program.

| Phase | The content of the program |
|---|---|
| Preparation period | 1.Understand the needs of preschool<br>2.Explain the content, implementation method and evaluation of the service-learning plan<br>3.Provide relevant information and professional knowledge to help students carrying out service activities |
| Action period | 1.Visit preschool and establish mutually beneficial partnerships<br>2.Perform eight storytelling activities for students |
| Evaluation period | 1.Share the results of service-learning for students<br>2.Discuss the process and experience of service-learning for students<br>3.Vote for the best service award for teacher and students |

and use, and storytelling skills, to undertake service activities in the form of eight storytelling sessions at preschools. The design of this service-learning program is based on Geleta and Gilliam [53], as shown in Table 1.

The researcher administered various pre-service, in-service, and post-service reflection activities to guide students in reflecting on the relationship between the service and the students themselves [19], the relationship between the service and classroom learning, the relationship between the service and the service recipients, and their thinking and personal gain after the service. Table 2 illustrates the design of reflection activities.

## 3.2 Participants

Participants in this study were undergraduate students enrolled in a Preschool Classroom Management (N = 36) course at a four-year institution in Taiwan (see Table 3). The course is considered introductory to understanding the concept and skill of Preschool Classroom Management from early childhood education perspective, and it is designated by the Ministry of Education as a Teacher Education Curriculum. Students in the course are typically in the third or fourth year of their undergraduate coursework.

The gender of the participants is mostly female, and their age is between 20 and 21. These students had only one-time service-learning experience before, and most of the service content

Table 2. Structured reflection activity design.

| Stage | Reflection Activity Type | Reflection Activity Content |
|---|---|---|
| Pre-service | Writing: preliminary service worksheet | 1. Guide students to reflect on the significance of the service personally for themselves<br>2. Describe the feelings toward and expectations of the service<br>3. Guide students to contemplate on the relationship between themselves and the service recipients and explore the effects of the service activity on themselves and on the learning activities of others. |
| In-service | Writing: eight self-reflection worksheets<br>Discussion: classroom discussion and sharing | 1. Guide students to describe the service activity.<br>2. Guide students to identify the relationship between the service activity and the course content.<br>3. Guide students to consider the significance, relation, and problems of the service activity in relation to themselves.<br>4. Guide students to contemplate how the service experience affected or changed them or their profession. |
| Post-service | Discussion: classroom group discussion and sharing | 1. Guide students to reflect on events that occurred in the course of the service as well as how the events mattered to themselves or others and encourage students to recollect what their service meant to them.<br>2. Students share their service experiences and personal gain. |

**Table 3. Participants' information (N = 36).**

| Items | Frequency | % |
|---|---|---|
| **Gender** | | |
| **Male** | 6 | 16.7 |
| **Female** | 30 | 83.3 |
| **Total** | 36 | 100 |
| **Age (years)** | | |
| **20** | 32 | 88.9 |
| **21** | 4 | 11.1 |
| **Total** | 36 | 100 |
| **Prior SL Experience** | | |
| **Yes** | 36 | 100 |
| **No** | 0 | 0 |
| **Total** | 36 | 100 |
| **Prior SL Setting** | | |
| **Library** | 16 | 44.4 |
| **Church** | 4 | 11.1 |
| **Tutoring centres** | 4 | 11.1 |
| **Charitable organizations** | 6 | 16.7 |
| **Daycare centre** | 6 | 16.7 |
| **Total** | 36 | 100 |

was clerical and cleaning work. The service organization was not in preschool and had no actual contact with children. This is their first experience in serving children.

## 3.3 Partner institution

The partner institution for this SL curriculum is Sunny Preschool (alias). Many of the institution's preschoolers were raised by grandparents (32%) or immigrant families (42%), and lacked reading opportunities. The preschool director wished to arrange activities that could improve the preschoolers' reading abilities. The researcher briefed the preschool director at the beginning of the semester on the storytelling service activity, which would not only allow preschoolers to experience storytelling but would also provide opportunities of classroom management to students who took the Preschool Classroom Management course. The preschool agreed to proceed this service activity, and thus the reciprocal SL curriculum was implemented accordingly. Throughout the semester, eight service activity sessions took place, with a storytelling activity taking place at 2-week intervals. The instructors of the preschool divided their respective classes into several groups, and each student conducted storytelling and extended activities on a group basis. The sessions were 1.5-hour-long each and scheduled on Wednesday afternoons.

## 3.4 Data collection and analysis

The data for the study was collected from September 2017 to February 2018, and data analysis was conducted for four months. The study was completed in June 2018. This article aims to expand the understanding of the experiences of students' reflections at various stages of the academic SL curriculum. Therefore, the case study method was adopted [54]. This study employed the participants' reflection documents as the primary data, supplemented by focus group interview records and research logs. Reflection documents refer to documents written by the participants during preservice, in-service, and postservice reflection activities, including

one worksheet for service preliminaries, eight reflection worksheets, and two in-class discussion records. Focus group interviews use interviews in a group context, through the process of group interaction and discussion, to achieve the purpose of collecting information [55]. Throughout the semester, 36 participants were divided into four focus group interviews. The interviews were scheduled after classes, for approximately 1.5 hours each. The interview outline was drafted based on the research questions. The focus group schedules (S1 Appendix) for the students were designed to provide participants with the opportunity to discuss their experience of service-learning and the impact on learning. Moreover, participants were invited to share thoughts or comments regarding the arrangement of reflection activities in the interviews.

This research took the thematic analysis approach to analyze and categorize the qualitative data [56]. In the data code S-P1, S represents the worksheet for service preliminaries; and P1 represents participant 1. In R-1-P3, R represents the reflection worksheet; P3 represents participant 3, and 1 represents the first reflection worksheet. In C-2-P1, C represents a in-class discussion records; P1 represents participant 1, and 2 represents the second in-class discussion records. In the data code F-P1, F represents a focus group interview; and P1 represents participant 1.

The data were first organized and then analyzed; in the data analysis, the researcher read the raw data closely and repetitively. The researcher first suspended their preperceptions and value judgments to let the data speak for itself and endeavored to uncover meanings in the data [57]. The researcher took a line-by-line analysis approach to data coding, assigned preliminary names through conceptualization, and used the constant comparative method to reveal the attributes and dimensions of concepts [58]. The researcher grouped similar ideas into categories and further categorized them into abstract concepts, which were then developed into the main concepts [57]. The process continues until no new concepts appear, that is, the data is saturated [55], and the core concepts are compiled at this stage [59]. Finally, the researcher produced research results based on the main concepts.

## 3.5 Reliability and validity

To increase the research reliability, the researcher performed triangulation [55], collecting data from various sources, namely participants' reflection worksheets, records of in-class discussions, and interview records to verify the results. In addition, the participant feedback technique was applied, which involved inviting research participants to validate the interview data collected and organized by the researcher to modify any erroneous data such that the original meaning could be preserved. The researcher described in detail the methods and strategies used to obtain the research data and also reflect on themselves constantly to minimize personal bias. The aforementioned verification strategies were intended to improve the reliability of the research.

## 3.6 Research ethics

Before the implementation of this research, the researcher explained to the participants that the purpose of the research was to provide opinions on the implementation of service-learning courses in higher education. In addition, we explained how the research was conducted. Ultimately, all 36 participants agreed to participate in the study.

For ensuring all participants' privacy and confidentiality, absolutely no study data can be identified with an individual participant. In addition, all data is anonymized before being used in research. Researchers reaffirmed the principle of confidentiality. Pseudonyms were assigned to all participants to protect their identity. We used 'F-P1', F represents a focus group

**Table 4. Perspectives of participants' reflection.**

| | |
|---|---|
| 1.Reflection Content and Performance for Different Stages | 1–1 Pre-service: Personal expectations |
| | 1–2 In-service: Professional learning and Personal growth |
| | 1–3 Post-service: Social concern |
| 2.Acquire Active Learning from Service Experience | |
| 3.Different Reflection Activities Trigger Participants' Different Reflection Dimensions and Functions | 3–1 Writing activities focus on personal reflection, allowing for self-examination and adjustment |
| | 3–2 Group discussion activity focused on dialogue, enabling learning and identity acquisition |
| 4.Students' Emotions Shift from Anxiety and Fear to Joy or Disquietude | |

interview; and P1 represents participant 1. In order to avoid physical and psychological harm to the participants due to the research, the researcher stated to the participants that if there is any uncomfortable feeling during the research process, please tell us immediately and we will consider the participants' wishes.

# 4. Results and discussion

In order to summarize the main themes and subthemes in Section 4, the Table 4 is used and briefly follows.

## 4.1 Participants' reflection content and qualitative change produced during the course of the service

The quality of reflection is more important to a student's educational achievement than the quantity of reflection [25]. Following a full semester of SL activities, the contents of participants' reflections at different stages of their service underwent a type of qualitative change, described in the following. Table 5 illustrates the reflection content and performance for different stages.

**4.1.1 Pre-service reflection contents focused on personal expectations.** As participants had not yet engaged in actual service actions, the pre-service reflection content predominantly emphasized personal expectations. These expectations include increased practical experience and good performance in storytelling. For example, P1 said: "I had no experience of storytelling. I know storytelling is a skill that teachers must have, and I hope to accumulate my experience of telling stories to children; that way, I will be able to perform well when I start working

**Table 5. Reflection content and performance for different stages.**

| Stage | Reflection Content | Performance |
|---|---|---|
| **Pre-service** | Personal expectations | 1. Increasing practical experience<br>2. Telling stories well |
| **In-Service** | Professional learning | 1. The skill of using vocal variations to attract the preschoolers' attention<br>2. Learning to choose the appropriate picture book<br>3. Noticing how a family environment affected children |
| | Personal growth | 1. Storytelling to be an experience of challenging oneself<br>2. Improving their ability to improvise<br>3. Bringing a sense of accomplishment |
| **Post-service** | Social concern | 1. Considering education to be the fundamental basis of the country<br>2. Pondering on the responsibilities of a teacher<br>3. Paying attention to the teaching and nurturing of children from disadvantaged groups |

in preschools" (S-P1). P14 said,". . .I really look forward to telling stories to children in pre-school! I will be well prepared and do my best so that children can listen to my stories and learn about caring for animals" (S-P14). Pre-service reflection encourages participants to set goals, and share their expectations of service-learning [15,19]. The present study found that the participants' reflection content focused on personal expectations, and such reflection indicated that participants were mentally prepared for the service activity [12], which was condu-cive to enhancing their engagement.

**4.1.2 In-service reflection content focused on professional learning and personal growth.** When participants began service activities, the content of their reflections shifted toward professional learning and personal growth. In terms of professional learning, P22 grasped the skill of using vocal variations to attract the preschoolers' attention. She said "When speaking to first-year preschoolers, one needs to slow down. Otherwise, they are unable to understand well, and different characters need to have different voices! I had to exaggerate my voice to draw their attention" (R-1-P22). P13 learned to choose the appropriate picture book for preschoolers. She said, "The picture book's content needs to be suitable for the spe-cific age range. The pictures need to be large, and the plot needs to be interesting enough to keep kids engaged. After these storytelling sessions, I learned that picture book selection was very important!" (F-P13).

Previously, students didn't know that preschoolers' behavior would be affected by their families. In the process of service-learning, this idea of students has changed. Students' inter-pretation and reinterpretation of events in order to make their own experiences meaningful is the key to understanding transforming in viewpoints and thinking habits [23]. P25 said, "After this experience, I found that children are not empty-headed. There was a kid who had a dog at home, and he knew how to take care of it. This may be related to the family environment" (R-3-P25).

These reflections are connected with the participants' professional knowledge. In agree-ment with previous studies [3,6,21,22,46], our data suggested that the service activity in ques-tion is combined with the Preschool Classroom Management course, providing the participants with the opportunity to integrate professional knowledge and service experience. Therefore, when instructors are designing service activities, the prior knowledge of students must be taken into account so that students may acquire professional learning through reflec-tion. Just as Eyler [18] observed, context is one of the critical elements contributing to effective SL reflection.

In terms of personal growth, the participants considered storytelling to be an experience of challenging oneself. P19 said: "The second time when I was in the first-year class, I expected that the kids would be unable to sit still, so I talked slowly and used simple sentences to tell the story. The children were really engaged! The best thing is that I succeeded in the challenge!" (F-P19). The storytelling experience allowed them to improve their ability to improvise. P12 said, "There were more disruptions than I had expected during the story-telling session for the children. Asking them to quieten down was not easy. I thought to myself that the hubbub could not go on like that, so I stopped talking. They started to be aware that they might have done something wrong and then settled down" (F-P12). Some participants thought that the service experience brought a sense of accomplishment. P7 said, "After finishing the storytelling session, I would think of the children's laughter and thought that I must have done something right [because] I can make them happy simply by telling them stories. I feel a great sense of accomplishment" (R-5-P7).

The service-learning experience creates a situation that challenges students' assumptions and perceptions and stereotypes of others, which leads to a sense of disequilibrium. This sense of disequilibrium requires reflection to help students clarify and critically evaluate personal

values, and develop values and attitudes based on new knowledge and experience [24]. This study is consistent with other research [2,20–22,52], the participants connected the service experience with their mental aspects and attained psychological growth by exploring their own capabilities.

**4.1.3 The content of post-service reflections focused on social concern.** After the service activities concluded, the participants had accumulated their experience of interacting with preschoolers and extended their reflection to include social issues. P11 considered education to be the fundamental basis of the country. Childhood is the stage worth the most investment, and hence to foster a decent individual, one must start from their childhood. This participant said, "Children are the future of the country. We should educate them well at a very early stage. Listening to stories is like taking in nutrition. Society can only become better if they can grow up happily" (F-P11). P15 pondered on the responsibilities of a teacher and said, "I know that being a teacher is not easy. Teachers must be responsible, caring, and patient. Teachers have a tremendous influence on children's futures and should thus teach them well" (C-2-P15).

According to P26, interacting with preschoolers who were being raised by grandparents, single parents, or immigrant parents prompted her to pay attention to the teaching and nurturing of children from disadvantaged groups. She pointed out the child-rearing challenges facing these families and indicated that these challenges ought to be alleviated by the government. P26 said "Parents who are busy working and single parents usually leave their children with grandparents, but parents should also make time for their children! Of course, the parents have to work and cannot do anything about it, and thus the government should provide assistance to these parents and lessen their pressure of child-rearing" (F-P26).

This study support research [3,6,20,21,48] that the participants began to take an interest in social issues and may be attributed to their experiences of interacting with the preschoolers, who motivated them to think social issues from their professional perspectives. The community of student life is different from the children, family and community they come into contact with in service learning [2]. Therefore, the experience of service-learning enables students to examine their own values and limited understanding of children, their families, and communities [60]. The impact on students is a commitment to social problems and social responsibilities [13]. These social concerns may subsequently be internalized as the participants' values, and this would help the participants not only to develop professional identities but also to constitute the core values for their future education career [2,6,48,61].

## 4.2 Acquire active learning from service experience

To the participants, storytelling in a preschool was an unfamiliar experience. Through service activities, the participants actively learned about aspects of classroom management, such as establishing classroom disciplines, formulating ground rules, and drawing children's attention. For example, P10 said, "Initially I thought that storytelling was about serving kids, but I later found out that I need to learn first how to draw their attention before starting telling a story. Also, the children must be fully informed of the classroom rules before a class started to avoid chaos" (F-P10). P3 said, "As I told stories to kids, I learned to change the volume of my voice to draw their attention. Also, they were asked to raise their hands before speaking. Only in this way could the storytelling go smoothly..." (C-2-P3). The participants' reflections were related to their service experiences.

According to the reflection-in-action concept of Argyris [62], service activities facilitate students' learning because reflection enables them to connect experience and learning; reflection-in-action is realized by actively reflecting on experience and constructing knowledge according to the experience. Furthermore, it is a teaching method that drives students to learn actively.

Service activities provided students with situated learning experiences. They gained experience of managing classes in a preschool, and such experience was conducive to constructing knowledge in situated contexts [63]. Moreover, the participants may apply the learned skills in future workplace, achieving learning transfer.

## 4.3 Different reflection activities trigger participants' different reflection dimensions and functions

**4.3.1 Writing activities focus on personal reflection, allowing for self-examination and adjustment.** The written reflection activities were a kind of self-monitoring [19], and were arranged in the course. The participants thought that writing reflection worksheets was similar to engaging in a dialogue with oneself. The process allows the participants to reflect on their strengths and weaknesses and the parts where adjustments could be made. P5 said, "Every time I write a reflection worksheet, I feel like I am talking to myself, asking what it was that I did well or badly? How might I adjust the next time? Why did the children react differently from how I expected?" (F-P5). P18 said, "I can track my progress as I write the worksheets: In the first session, there was unorganized, the children were not focused, and the storytelling did not go smoothly. The second time, the class was well organized, and I had better storytelling skills" (F-P18).

These responses clearly indicated that writing facilitated a personal, internal mental process that allowed the participants to describe in detail their reflections on their actions, to question their actions in the process, and to identify solutions to problems [64]. Hsu [65] suggested that a self-reflection constitutes an in-depth dialogue between oneself and data. The participants collected data from their actions and questioned their own ideas, allowing them to achieve a more profound understanding.

**4.3.2 Group discussion activity focused on dialogue, enabling learning and identity acquisition.** Focus group inquiry as a reflective practical experience provides participants with an opportunity to discuss their experiences and learn about these experiences with their colleagues [13]. Group discussions were incorporated into the curriculum. The participants suggested that they were able to learn from their classmates and acquire identity through group dialogues. This study supported previous research [5,23] that dialogue contexts triggered the participants' reflection on their behavior and relationships with others, which in turn made them analyze themselves through a metacognitive approach and enabled the production of new knowledge. P17 said, "From what was shared, I found that many classmates were initially anxious and reserved and that I was not the only one feeling this way. My classmates also talked about how to overcome anxiety. Maybe I can give this method a try" (R-6-P17). P6 said, "Everyone had experiences with children interrupting or pushing others around! One of my classmates said that she was not sure what to do about it at that time. Some classmates said that the storytelling should stop, which would draw the children's attention. I used this approach as well" (R-4-P16).

Dialogue or discussion is a form of reflection [19]. Through dialogue between group members, participants can conduct rational communication and reach cognitive consensus. Input from others can also prompt reflection from multiple perspectives, new platform for problem-solving, connecting theory to practice, and provide emotional support [33,66], and provide participants with opportunities for explicit tacit knowledge [13].

## 4.4 Students' emotions shift from anxiety and fear to joy or disquietude

The current discourse on the SL model focused more on the procedural execution of the project participants, with less consideration given to the participants' internal mental states [67].

The mental state of students participating in SL is critical. The present study showed that students participating in the service process experienced different emotional changes: Initial anxiety and fear turned to joy or disquietude. At first, the participants thought the only thing they had to do was telling stories. However, they became nervous when they entered the classroom. P29 said, "This is my first time coming to a preschool and being a storyteller, so I have that felt unsettled as a novice" (R-1-P29). Some students feared unpredictability. P21 said, "Everyone felt nervous before they started the stories, fearing that children would either go wild, become disinterested, or simply not understand the story" (R-1-P21). Students had high expectations before beginning their service, but once they entered the classroom, they experienced anxiety and fear. These emotions occurred probably as a result of their first engagement in such service experience.

Students' emotions began to change as their storytelling experiences accumulated. They started to experience a sense of joy through the sense of accomplishment gained from the preschoolers' smiles, focused gazes, eager opinions and responses to questions, and compliance to rules. P20 said, "When children give eager responses and focus on storytelling, one sees the smiles on their faces and feels an indescribable sense of joy and accomplishment" (R-5-P20). A person successfully copes with distressing dilemmas and/or urgency learning as transformative learning [68]. However, some students experienced negative emotions. P4 encountered an unexpected and unmanageable quarrel among the preschoolers during a storytelling session. A sense of disquietude developed. She said, "I did not expect a quarrel among the kids. The situation got a bit out of hand. I could not proceed the story. That left me at a loss as to what to do" (C-2-P4). As students' new experiences do not match their expectations and cause frustration or anxiety, they may rethink their new information and experiences or change how they think [19,40], thereby improve their transformative learning [23].

## 5. Conclusions and recommendations

### 5.1 Conclusions

The purpose of this research is to explore the reflection content of 36 college juniors participating in academic SL courses. Based on the qualitative research orientation, the focus group interviews and the collection of student reflection documents are used for research. The important findings of this study are as follows: The participants had different reflections at the pre-service, in-service, and post-service stages, undergoing qualitative changes as their service experience accumulated. At the pre-service stage, the reflection content focused on personal expectations, which then shifted to professional learning and personal growth at the in-service stage. Post-service reflections were extended to social concern. The participants were able to actively acquire professional knowledge through their experience of service. Different reflection activities triggered different reflection levels and functions among the participants. Writing activities caused the participants to focus on personal reflection, providing them an opportunity to review and adjust their approaches. Group discussion activities allowed students to engage in dialogues, learn from others, and acquire identity. At the in-service stage, students' emotions fluctuated considerably, switching from nervousness and fear to joy or a sense of disquietude. The findings of this study proved that service-learning can promote students' professional learning, personal development and social concern. This part is consistent with the previously reported results [5–7,14,21]. The implications for higher education in Taiwan, service learning provides college students with a teaching method that connects theory and practice, and it is continuous experiential learning rather than a one-time experience. This kind of experiential learning experience can help students to understand and rethink their role

as community members, and critically think and put forward opinions on social issues from their own professional perspective.

## 5.2 Recommendations

According to the research results, the following suggestions are provided. The suggestion to university teachers is that instructors may design structured reflection activities for the SL curriculum with reference to the various service stages, and provide continuous reflection activities for students throughout the semester. When designing reflection activities, instructors might take into account students' learning preferences to design reflection activities, such as writing or discussion activities on an individual or a group basis, and accommodate various students' reflection needs. Instructors should design reflection activities that provide guidance and take into account students' emotional states. Moreover, appropriate support and feedback should be available to assist students in learning how to address negative emotions during the service process. Suggestions for students, students should actively participate in academic SL courses, accumulate practical experience through "learning in service" and "service in learning", revise and adjust teaching experience from "reflection on action". When students have more opportunities to engage in service-learning, the more likely they are to learn to overcome obstacles.

## Supporting information

**S1 Appendix.**
(DOC)

## Author Contributions

**Conceptualization:** Ting-Hua Lin.

**Data curation:** Ting-Hua Lin.

**Formal analysis:** Ting-Hua Lin.

**Funding acquisition:** Ting-Hua Lin.

**Investigation:** Ting-Hua Lin.

**Methodology:** Ting-Hua Lin.

**Project administration:** Ting-Hua Lin.

**Resources:** Ting-Hua Lin.

**Software:** Ting-Hua Lin.

**Supervision:** Ting-Hua Lin.

**Validation:** Ting-Hua Lin.

**Visualization:** Ting-Hua Lin.

**Writing – original draft:** Ting-Hua Lin.

**Writing – review & editing:** Ting-Hua Lin.

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
