## [Decision Letter · Decision Letter 0]

14 Jan 2021

PONE-D-20-32439

Revelations of Service-Learning Project:

Multiple Perspectives of College Students’ Reflection

PLOS ONE

Dear Dr. Lin,

Thank you for submitting your manuscript to PLOS ONE. After careful consideration, we feel that it has merit but does not fully meet PLOS ONE’s publication criteria as it currently stands. Therefore, we invite you to submit a revised version of the manuscript that addresses the points raised during the review process.

The two reviewers both acknowledged that this is an interesting topic and the author has done a good job in using the data to support the argument. However, they also raised a series of concerns which are expected to be resolved through an extensive revision. 

We look forward to receiving your revised manuscript.

Kind regards,

Mingming Zhou, Ph.D.

Academic Editor

PLOS ONE

Journal Requirements:

Reviewers' comments:

Reviewer's Responses to Questions

**Comments to the Author**

1. Is the manuscript technically sound, and do the data support the conclusions?

Reviewer #1: Yes

Reviewer #2: Yes

2. Has the statistical analysis been performed appropriately and rigorously? 

Reviewer #1: N/A

Reviewer #2: N/A

3. Have the authors made all data underlying the findings in their manuscript fully available?

Reviewer #1: Yes

Reviewer #2: Yes

4. Is the manuscript presented in an intelligible fashion and written in standard English?

Reviewer #1: Yes

Reviewer #2: Yes

5. Review Comments to the Author

Reviewer #1: General comments

- The article is interesting, the topic is actual and the text is well-written, however, from my point of view further improvements should be made. I strongly encourage the author to continue working on it because the topic is relevant and highly topical. It is important to build a rigorous corpus of knowledge based on empirical evidence that supports all the research that is being done in this regard. Therefore, several suggestions should be made in order to keep improving it and achieve the high-quality level required by this journal.

- I recommend to the authors the option of linking the expression ‘service-learning’ with a script, since there is scientific-pedagogical literature that advises it. In my opinion it is a convenient option since it implies an indissoluble union between the terms that best defines the global concept of SL. In any case it is a personal option but, please, authors should be consistent in the entire text.

- Please, author should make sure that the text follows the journal guidelines (formatting, references, etc.)

Title:

- In my opinion, the current title fits the content of the manuscript.

Abstract:

- In the current version of the abstract, the method is not yet clearly explained. For instance, I would appreciate to know clearer information on the service actions on which the SL program was built. Finally, I strongly advise for a proper specification/description and justification of the data analysis approach and techniques.

Introduction

- It would be essential to clarify the gap existent in this field in order to justify the reasons why it is necessary to investigate along these lines. There are several studies on SL in the field of Physical Education Teacher Training that could update and reinforce this foundation on SL reflection outcomes.

- The aim of the study should be clearly specified and justified.

I would like the author to reinforce the final paragraph in page 4 with more references.

Method

- I am surprised that all the participants had previous SL experience. Could this background have conditioned the results somehow? Could you deepen on this issue?

- In relation to the SL intervention programme carried out, it should be better organized from my point of view

- On what theoretical bases was the structure of the focus group interviews supported?

- Could you better explain the sampling, analysis and saturation processes? In this vein, I think it would be necessary to incorporate references to strengthen the analysis section: qualitative approach, saturation, sampling techniques…

Results and Discussion

- If possible, the results and discussion section should be completed with more recent and specific references on SL in teacher training.

- There are a lot of subsections in this section. Is it possible to simplify or clarify them?

Conclusion:

- I would appreciate the incorporation of the main limitations and future research lines of the study.

Final comments

I would like to encourage the authors to work in the suggested comments, since the topic and the results obtained seem truly relevant and of great interest to me. If you have any other questions do not hesitate to contact and ask for as many explanations as you need.

Reviewer #2: This paper is quite interesting and delves into SL experiences in undergraduate students working with children, which is a topic of outmost importance. In order to improve the quality of the paper, here are my recommendations:

1. The introduction gives context and highlights important features of service learning. However, I think it is too specific when describing the theoretical background using Dewey and Kolb’s theories. Instead of that, I would include more information on the context of service learning in Taiwan, particularly with pre-schoolers.

2. It is not clear if the research had a Research Ethics Committee approval. It would be worth mentioning if it did and, if not, why did the researcher decide not to have this approval. This is particularly important, as apparently the professor in charge of the course is also the researcher. What are the implications in terms of the pre, during and post activities and data collection strategies?

3. Table 3 is not clear. Why does it start with 5 and 6 in the first row?

4. Results can be presented at the beginning in a table summarising the main themes and subthemes or in a figure. Sometimes it was hard to follow.

5. The results section can benefit of more in-depth analysis. Results could be more specific in terms of the transformation path of the students, and further explain themes such as personal growth, or acquire active learning, for example. These are some of the examples:

a. Personal growth is very descriptive and doesn’t provide an analysis of all the possible links to the student’s responses. The first stage (pre-service) is definitely the moment to set personal expectations, however, it would be worth noting the different types of expectations, if there were individual only, or focused on the collective, for example.

b. In the case of “in service reflection…” theme, this particular part: “After interacting with the preschoolers, the participants noticed how a family environment affected children. P25 said, “After this experience, I found that children are not empty-headed. There was a kid who had a dog at home, and he knew how to take care of it. This may be related to the family environment” (R-3-P25), is an example of context in the SL experience, however it shouldn’t be the only one because this is one of the key aspects of the SL strategy. Later on, during post-service reflections, you mention how students are much more aware of the challenges at the societal level, this is a theme that is worth exploring with more depth to really analyse how the students move from recognising children as thinking humans, to understanding the complexities of the social sphere.

6. Conclusions present central issues developed throughout the paper. However, I would suggest to link it much more with the national context, the implications for undergraduate teaching and how this pedagogical strategy serves to improve societal issues, for example.

7. Recommendations can be more specific on specific actions to be taken at different levels.

6. PLOS authors have the option to publish the peer review history of their article (what does this mean?). If published, this will include your full peer review and any attached files.

Reviewer #1: No

Reviewer #2: **Yes: **Laura Fonseca

---

## [Decision Letter · Decision Letter 1]

14 May 2021

PONE-D-20-32439R1

Revelations of Service-Learning Project:

Multiple Perspectives of College Students’ Reflection

PLOS ONE

Dear Dr. Lin,

Thank you for submitting your manuscript to PLOS ONE. After careful consideration, we feel that it has merit but does not fully meet PLOS ONE’s publication criteria as it currently stands. Therefore, we invite you to submit a revised version of the manuscript that addresses the points raised during the review process. 

We look forward to receiving your revised manuscript.

Kind regards,

Mingming Zhou, Ph.D.

Academic Editor

PLOS ONE

Journal Requirements:

Reviewers' comments:

Reviewer's Responses to Questions

**Comments to the Author**

1. If the authors have adequately addressed your comments raised in a previous round of review and you feel that this manuscript is now acceptable for publication, you may indicate that here to bypass the “Comments to the Author” section, enter your conflict of interest statement in the “Confidential to Editor” section, and submit your "Accept" recommendation.

Reviewer #1: (No Response)

2. Is the manuscript technically sound, and do the data support the conclusions?

Reviewer #1: Yes

3. Has the statistical analysis been performed appropriately and rigorously? 

Reviewer #1: N/A

4. Have the authors made all data underlying the findings in their manuscript fully available?

Reviewer #1: Yes

5. Is the manuscript presented in an intelligible fashion and written in standard English?

Reviewer #1: Yes

6. Review Comments to the Author

Reviewer #1: The article has improved substantially since the last revision I did. In my opinion, the current version of the manuscript almost meets all the elements required for its publication in such hight impact journal. I would only ask the authors that, at this point of the process, it would be convenient to update and reinforece the theoretical framework and discussion section with recent articles (published during the last three or less years) with maximum impact on this topic. The following are suggested. All of them present common features that could slightly enrich the authors approach.

- Marttinen, R., N. D. Daum, D. Banville, and R. N. Fredrick. (2020). “Pre-service teachers learning through service-learning in a low SES school”. Physical Education and Sport Pedagogy, 25 (1): 1-15.

- Chiva-Bartoll, O., Moliner, L., Salvador-García, C. (2020). Can service-learning promote social well-being in primary education students? A mixed method approach. Children and Youth Services Review; 111 pp. 1-8.

- García-Rico, L., Martínez-Muñoz, F., Santos-Pastor, M.L., Chiva-Bartoll, O. (2021). Service-learning in physical education teacher education: a pedagogical model towards sustainable development goals (Epub ahead of print). International Journal of Sustainability in Higher Education; online pp. --.

- MacPhail, A., and R. Sohun . (2019). “Interrogating the enactment of a service-learning course in a Physical Education teacher education programme: Less is more?”, European Physical Education Review 25 (3): 876-892.

7. PLOS authors have the option to publish the peer review history of their article (what does this mean?). If published, this will include your full peer review and any attached files.

Reviewer #1: No

---

## [Author Response · Author response to Decision Letter 1]

30 May 2021

PONE-D-20-32439：Response to Reviewers

We would like to thank the reviewers for their time and efforts reading our manuscript and providing insightful suggestions for the improvement of our work. We have read the comments carefully, and tried to follow the suggestions as closely as possible. Specific changes in this revision are summarized below.

For Reviewer #1

Q1:

The article has improved substantially since the last revision I did. In my opinion, the current version of the manuscript almost meets all the elements required for its publication in such hight impact journal. I would only ask the authors that, at this point of the process, it would be convenient to update and reinforece the theoretical framework and discussion section with recent articles (published during the last three or less years) with maximum impact on this topic. The following are suggested. All of them present common features that could slightly enrich the authors approach.

- Marttinen, R., N. D. Daum, D. Banville, and R. N. Fredrick. (2020). “Pre-service teachers learning through service-learning in a low SES school”. Physical Education and Sport Pedagogy, 25 (1): 1-15. 

- Chiva-Bartoll, O., Moliner, L., Salvador-García, C. (2020). Can service-learning promote social well-being in primary education students? A mixed method approach. Children and Youth Services Review; 111 pp. 1-8.

- García-Rico, L., Martínez-Muñoz, F., Santos-Pastor, M.L., Chiva-Bartoll, O. (2021). Service-learning in physical education teacher education: a pedagogical model towards sustainable development goals (Epub ahead of print).International Journal of Sustainability in Higher Education; online pp.--

- MacPhail, A., and R. Sohun. (2019). “Interrogating the enactment of a service-learning course in a Physical Education teacher education programme: Less is more?”, European Physical Education Review 25 (3): 876-892.

A1:

Thank you for your comment. 

A1-1:

This paper "Pre-service teachers learning through service-learning in a low SES school" proposed by Marttinen, R., ND Daum, D. Banville, and RN Fredrick (2020) has been published on page 6 of this article, added in p .14, as shown in highlight.

 (a) Professional learning: Improve student understanding and practical capabilities in relation to adaptive education [5]; enhance students’ pedagogical knowledge, capabilities of planning and implementing collaborative curriculum, skills of designing crossdisciplinary pedagogical activities, and professional attitudes [7]; help students develop curriculum-design and pedagogical capabilities, gain greater understanding of pedagogical events, maintain favorable instructor–student relationships, develop counseling abilities, enhance management skills, improve interaction skills with instructors, and foster ethical professional values and attitudes [3]; enhance the self-efficacy of student’ teaching technology, the belief in the integration of teaching technology, and understand the role of teaching technology in education [43 44]; development of professional identity of pre-service teachers [44 45]; and developing classroom and behavior management skills [45 46].

These reflections are connected with the participants’ professional knowledge. In agreement with previous studies [3, 6, 21, 22, 46], our data suggested that the service activity in question is combined with the Preschool Classroom Management course, providing the participants with the opportunity to integrate professional knowledge and service experience. Therefore, when instructors are designing service activities, the prior knowledge of students must be taken into account so that students may acquire professional learning through reflection. Just as Eyler [18] observed, context is one of the critical elements contributing to effective SL reflection. 

A1-2:

The research participants of this article is college students, but the research participants of the paper “Can service-learning promote social well-being in primary education students? A mixed method approach” of Chiva-Bartoll et al. are primary education students (third to fifth grade), so citation is not considered.

A1-3:

This paper "Service-learning in physical education teacher education: a pedagogical model towards sustainable development goals" proposed by García-Rico et al. (2021) has been supplemented on p.6, p.7, p.15, as shown in highlight..

(b) Individual Growth: Boost students’ self-worth [46 47]; promote personal growth and career exploration [20, 48]; and guide students to perceive positively, identify personal weaknesses, hone communication skills and problem solving abilities, stimulate ideas through peer observation and collaboration, provide collective emotional support, learn to care for others, explore their career, stimulate service dedication, improvise actions according to immediate circumstances, and permit self-reflection [7, 21].

(c) Social Concern: Improve students’ sense of social responsibility and civic engagement [20, 47, 48, 49]; nurture students’ humanistic and social concerns and allow students to experience ethnic differences and care for the disadvantaged [3, 48]; make students aware of factors affecting preschool education quality [21]; provide students with information regarding social attitudes, leadership skill development, and opinions in the community [48 50]; enhance students’ understanding of the importance of community involvement [49 51]; and how they can contribute to the community [50 52].

This study support research [3, 6, 20, 21, 48] that the participants began to take an interest in social issues and may be attributed to their experiences of interacting with the preschoolers, who motivated them to think social issues from their professional perspectives. The community of student life is different from the children, family and community they come into contact with in service learning [2]. Therefore, the experience of service-learning enables students to examine their own values and limited understanding of children, their families, and communities [58 60]. The impact on students is a commitment to social problems and social responsibilities [13]. These social concerns may subsequently be internalized as the participants’ values, and this would help the participants not only to develop professional identities but also to constitute the core values for their future education career [2, 6, 48, 59 61].

A1-4:

This paper “Interrogating the enactment of a service-learning course in a Physical Education teacher education programme: Less is more?” proposed by MacPhail, A., and R. Sohun. (2019) has been supplemented on p.3, as shown in highlight.

Institutions of higher education across the world have begun actively promoting the SL curriculum in light of the striking results that the United States has achieved in implementing service-learning [27]. Kendall [28] remarked that service-learning has been perceived as a scheme, a philosophy, or pedagogy. SL is a type of pedagogy that achieves an instruction goal, namely connecting community services with educational institutions [6, 23, 29, 30]. It is currently considered a high-impact educational practice [30] [31].

---

## [Decision Letter · Decision Letter 2]

10 Sep 2021

Revelations of Service-Learning Project:

Multiple Perspectives of College Students’ Reflection

PONE-D-20-32439R2

Dear Dr. Lin,

We’re pleased to inform you that your manuscript has been judged scientifically suitable for publication and will be formally accepted for publication once it meets all outstanding technical requirements.

Kind regards,

Mingming Zhou, Ph.D.

Academic Editor

PLOS ONE

Additional Editor Comments (optional):

Reviewers' comments:

Reviewer's Responses to Questions

**Comments to the Author**

1. If the authors have adequately addressed your comments raised in a previous round of review and you feel that this manuscript is now acceptable for publication, you may indicate that here to bypass the “Comments to the Author” section, enter your conflict of interest statement in the “Confidential to Editor” section, and submit your "Accept" recommendation.

Reviewer #1: All comments have been addressed

2. Is the manuscript technically sound, and do the data support the conclusions?

Reviewer #1: (No Response)

3. Has the statistical analysis been performed appropriately and rigorously? 

Reviewer #1: (No Response)

4. Have the authors made all data underlying the findings in their manuscript fully available?

Reviewer #1: (No Response)

5. Is the manuscript presented in an intelligible fashion and written in standard English?

Reviewer #1: (No Response)

6. Review Comments to the Author

Reviewer #1: (No Response)

7. PLOS authors have the option to publish the peer review history of their article (what does this mean?). If published, this will include your full peer review and any attached files.

Reviewer #1: No

---

## [Editor Report · Acceptance letter]

14 Sep 2021

PONE-D-20-32439R2 

Revelations of Service-Learning Project:
Multiple Perspectives of College Students’ Reflection 

Dear Dr. Lin:

I'm pleased to inform you that your manuscript has been deemed suitable for publication in PLOS ONE. Congratulations! Your manuscript is now with our production department. 

Kind regards, 

on behalf of

Dr. Mingming Zhou 

Academic Editor

PLOS ONE